# SARS-CoV-2 infection in central North Carolina: Protocol for a population-based longitudinal cohort study and preliminary participant results

Elyse M. Miller[1]☯, Elle A. Law[1]☯, Rawan Ajeen[1], Jaclyn Karasik[1], Carmen Mendoza[1], Haley Abernathy[2], Haley Garrett[1], Elise King[2], John Wallace[3], Michael Zelek[4], Jessie K. Edwards[1], Khou Xiong[1], Cherese Beatty[1], Aaron T. Fleischauer[5,6], Emily J. Ciccone[7], Bonnie E. Shook-Sa[8], Allison E. Aiello[1,9], Ross M. Boyce[1,7]*

1 Department of Epidemiology, Gillings School of Global Public Health, University of North Carolina at Chapel Hill, Chapel Hill, NC, United States of America, 2 Institute of Global Health and Infectious Diseases, University of North Carolina at Chapel Hill, Chapel Hill, NC, United States of America, 3 North Carolina Institute of Public Health, Gillings School of Global Public Health, University of North Carolina at Chapel Hill, Chapel Hill, NC, United States of America, 4 Chatham County Public Health Department, Pittsboro, NC, United States of America, 5 Division of State and Local Readiness, United States Centers for Disease Control and Prevention, Atlanta, GA, United States of America, 6 Epidemiology Branch, North Carolina Department of Health and Human Services, Raleigh, NC, United States of America, 7 Division of Infectious Diseases, School of Medicine, University of North Carolina at Chapel Hill, Chapel Hill, NC, United States of America, 8 Department of Biostatistics, Gillings School of Global Public Health, University of North Carolina at Chapel Hill, Chapel Hill, NC, United States of America, 9 Carolina Population Center, University of North Carolina at Chapel Hill, Chapel Hill, NC, United States of America

☯ These authors contributed equally to this work.
* roboyce@med.unc.edu

**Funding:** This work was supported by a contract between the University of North Carolina at Chapel Hill and the North Carolina Department of Health

## Abstract

Public health surveillance systems likely underestimate the true prevalence and incidence of SARS-CoV-2 infection due to limited access to testing and the high proportion of subclinical infections in community-based settings. This ongoing prospective, observational study aimed to generate accurate estimates of the prevalence and incidence of, and risk factors for, SARS-CoV-2 infection among residents of a central North Carolina county. From this cohort, we collected survey data and nasal swabs every two weeks and venous blood specimens every month. Nasal swabs were tested for the presence of SARS-CoV-2 virus (evidence of active infection), and serum specimens for SARS-CoV-2-specific antibodies (evidence of prior infection). As of June 23, 2021, we have enrolled a total of 153 participants from a county with an estimated 76,285 total residents. The anticipated study duration is at least 24 months, pending the evolution of the pandemic. Study data are being shared on a monthly basis with North Carolina state health authorities and future analyses aim to compare study data to state-wide metrics over time. Overall, the use of a probability-based sampling design and a well-characterized cohort will enable collection of critical data that can be used in planning and policy decisions for North Carolina and may be informative for other states with similar demographic characteristics.

and Human Services, Division of Public Health (https://publichealth.nc.gov/), contract 00041877 awarded to author RB. The funders did not have and will not have a role in study design, data collection and analysis, decision to publish, or preparation of the manuscript.

**Competing interests:** The authors have declared that no competing interests exist.

## Introduction

In addition to the direct health impacts, the COVID-19 pandemic has caused unprecedented levels of disruption to the global economy and civil society. While critical to limiting disease transmission and associated morbidity and mortality, prevention measures have taken a significant toll [1, 2]. Nearly every aspect of daily life, including business, education, organized religion, and social activities, has experienced restrictions and temporary closures as a result of the pandemic.

Decisions regarding how and when to scale back such restrictions are complex. Premature easing may result in a rebound of cases [1] even in the presence of vaccines [3, 4], while extending restrictions may inflict irreversible damage to the economy and to children's health and development [5, 6], especially in already distressed rural communities. Until vaccination rates or "natural immunity" from exposure reach critical thresholds, guidance on the scope and duration of restrictions will continue to require epidemiological measurements of community infections.

Current estimates of SARS-CoV-2 incidence, prevalence across geographic regions, and mortality rates are largely drawn from seroprevalence studies [7], which measure antibodies against the virus found in blood samples. These studies vary by design, serological test employed, and statistical methods. In addition, a large proportion of studies to date have used convenience samples that may reflect very different populations and are subject to a number of biases, foremost of which is related to the selection of participants. Yet the results of these studies are frequently extrapolated to the general population and are interpreted interchangeably, despite not reflecting the underlying population in demographic composition and risk factors for COVID-19 infection, leading to estimates of questionable accuracy [8].

The seroprevalence of SARS-CoV-2 is rapidly changing in North Carolina [9], a state with fast-growing urban centers interspersed among the second-largest rural population in the country. Limited studies have been conducted among frontline health care workers [10] and among those seeking healthcare unrelated to COVID-19 [11]. However, no studies to date have used representative population estimation methods and often rely on convenience sampling, which is difficult to extrapolate to underlying populations [7]. Therefore, there is an urgent need to conduct prospective, population-based surveillance to define the epidemiologic curve and provide accurate and timely information to policymakers. This need is particularly acute as the Centers for Disease Control and Prevention (CDC) estimates that up to 70% of individuals infected with SARS-CoV-2 are asymptomatic [12], and others experience only mild symptoms that do not prompt care seeking and diagnostic testing [13, 14]. As the pandemic evolves and vaccination efforts expand in North Carolina [15], such surveillance also enables estimates of vaccine intention and uptake.

Here, we describe the protocol of an ongoing study that was designed to estimate and examine a truer population-based incidence and prevalence of SARS-CoV-2 infection in a representative sample of adults residing in one county in central North Carolina. We hypothesized that a population-based community incidence and prevalence estimate would be substantially higher than estimates derived from facility-based samples, largely due to limited access to testing during the early phase of the pandemic and the high proportion of infections that are asymptomatic or mild, and thus do not prompt care seeking. This observational study also sought to identify demographic, socioeconomic, and geographic risk factors for SARS-CoV-2 infection, and to characterize self-reported symptoms, health-seeking behaviors, and clinical outcomes in relationship to seroprevalence results.

## Materials and methods

### Study overview

We conducted a prospective, observational study of SARS-CoV-2 among residents of Chatham County, North Carolina (S1 Fig). Chatham County was selected as the setting because it has many characteristics that make it broadly representative of the state of North Carolina, including distinct semi-urban and rural areas and a diverse mix of residents both in terms of demographics and socioeconomic status (see S1 Table). The county was affected by high-profile outbreaks at nursing homes as well as poultry processing plants, which have disproportionately impacted Hispanics/Latinos and other historically marginalized communities [1]. Based on precision-based sample size calculations made early in the course of the pandemic, we aimed to enroll up to 300 participants. Eligibility criteria included current residence in Chatham County, age of 18 years or older, and willingness and ability to provide informed consent. As described in detail below, study participants were regularly surveyed on their demographic characteristics, occupation, infection prevention activities and behaviors, illnesses and health, and psychosocial health. Survey data on the health of their household members was also collected. Venous blood was collected from participants on a monthly basis, and nasal swabs were self-collected on a bi-weekly basis (Fig 1).

### Ethical considerations

Study protocols were reviewed by the University of North Carolina at Chapel Hill Institutional Review Board. Initial approval (Study #20–1632) was granted under expedited review

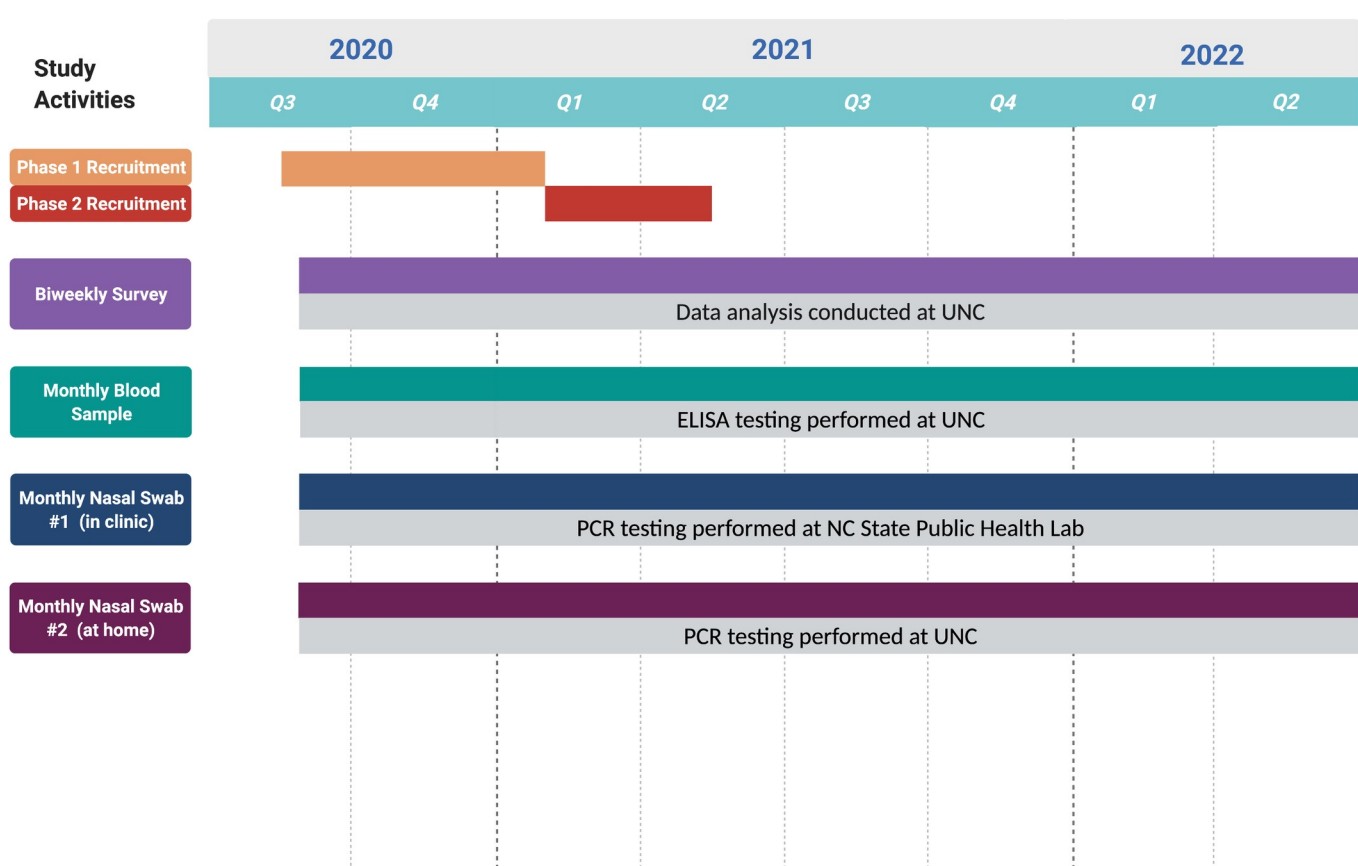

**Fig 1. Overview of study timeline.** Created with BioRender.com.

under 45 CFE 46.110 on June 10, 2020. Written informed consent was obtained from all participants.

## Recruitment

Initial study recruitment leveraged and expanded upon the Chatham County Community Cohort, which was established by the Chatham County Health Department during their 2018 Community Assessment to track the health status of a representative sample of residents over time [16]. Investigators met with the Chatham County Community Assessment Scientific Advisory Committee on April 9, 2020 to describe the study objectives and methods. Ample time was provided for committee members to ask questions and provide feedback, which was incorporated into the proposal.

To raise community awareness of the study, we collaborated with the Office of Rural Initiatives at the University of North Carolina at Chapel Hill (UNC). Informational flyers were handed out at food distribution sites and other community events in August and September 2020 and at mobile markets (drive-thru food distribution sites) in November and December 2020.

Recruitment efforts sought to enroll one person per sampled household. Participants were selected in two different phases. Phase One was recruited from the existing, population-based Chatham County Community Cohort. Residents of Chatham County were sampled for the Chatham County Community Cohort via a stratified two-stage cluster design. Census blocks were stratified into three income tertiles and 21 census blocks were selected per stratum using probability proportional to size with replacement (PPS-WR) sampling with the estimated number of occupied households in each census block serving as the measure of size. Income tertiles used the 33rd and 66th percentiles, so that income ranges used were: <$47,000; $47,000 - $58,000; and >$58,000. Within each selected census block, seven households were selected for inclusion in the Chatham County Community Cohort using a modified Community Assessment for Public Health Emergency Response (CASPER) sampling methodology [17]. Community Cohort participants who responded to a questionnaire given as part of the 2020 Chatham County Community Health Assessment via email or phone were offered the opportunity to participate in our study.

Phase One recruitment included contacting potential participants from the Chatham County Community Cohort via email and telephone, with up to three emails per email address and up to five telephone attempts per phone number. Recruitment among the 189 cohort members who responded to the 2020 questionnaire (Phase One) was not expected to yield the targeted number of participants for the study. Therefore, further recruitment was conducted within an additional sample of households, herein Phase Two. Our Phase Two design used the same sampling approach for selecting census blocks as the Chatham County Community Cohort, enabling us to combine samples from both phases. Namely, a stratified PPS-WR sample of 99 census blocks was selected for Phase Two. Address-based sampling frames were compiled for the selection of households from within the chosen census blocks [18], with inclusion of addresses from the supplemental No-Stat file to maximize sample representation in rural areas [19]. Within-block sample sizes were derived based on the demographics of selected blocks, with oversampling of addresses within census blocks with higher concentrations of Hispanic/Latino and/or Black/African American populations. This led to an overall supplemental sample of 1,402 addresses from 80 unique census blocks. Assuming a household occupancy rate of 90% and a response rate of 20%, the combined Phase One and Phase Two samples were anticipated to yield approximately 300 study participants.

Phase Two recruitment involved sending a postcard (S1 Fig) to households in our sample to provide information regarding the study and encourage enrollment. Postcards were sent up

to three times per address and referenced a study-specific website where members of sampled households could learn more about the study and submit an online pre-screening survey conveying their desire to be contacted for participation. Phone numbers were available for approximately 76% of the Phase Two addresses. As in Phase One, sampled households with available phone numbers in Phase Two were contacted by telephone up to five times. Study team members made in-person visits to selected Phase Two households that were not reachable by telephone and had not responded to postcards, beginning in February 2021. Staff wore identification and personal protective equipment and maintained at least six feet of distance from household members during visits. If there was no response, a flyer containing contact information for the study was left at the front door (See S2 Fig).

## Consent and enrollment

If a member of a household sampled in either Phase One or Phase Two expressed an interest in participation, staff conducted a verbal consent process with an electronic consent form using the AdobeSign program. A copy of the signed consent form was then provided to the participant by email. For participants lacking access to email, verbal consent was obtained over the phone then confirmed in writing at the participant's first in-person study visit.

Once enrolled, each participant was assigned a unique study ID for linkage of data sources.

## Questionnaire and surveys

**Baseline.** After enrollment, participants received an electronic questionnaire regarding their demographic characteristics, occupations, infection prevention practices, health history, current COVID-19 symptoms, and personal and professional impacts of the COVID-19 pandemic (S1 Appendix). All surveys were available in English and Spanish. This initial questionnaire also inquired about the participant's household (i.e., number of people, occupations, and current infection symptoms) and included basic mental health assessments for depression and anxiety [20, 21].

**Biweekly.** Participants received a shorter survey by email every two weeks to gather data on current occupational status, current infection prevention practices, personal and professional impacts of the COVID-19 pandemic, and symptoms experienced during the previous two weeks (S2 Appendix). The biweekly survey was also designed to collect information on household members, including current occupational status and symptoms experienced during the previous two weeks. In December 2020, the first COVID-19 vaccines were approved by the U.S. Food and Drug Administration and became available to the state of North Carolina [15]. To capture subsequent vaccination trends among participants, surveys were updated on March 11, 2021 with questions on when and if participants had received one or both vaccine doses, intention to vaccinate when available, and type of vaccine received.

## Specimen collection

Participants were given the option to provide specimens during monthly study visits with at-home specimen collection between visits or, if unable to attend study visits due to lack of transportation or other barriers, to contribute specimens entirely from home via self-collection (See Fig 2).

**Study visits.** Study visits occurred at one of two clinic sites in Chatham County. Participants attended monthly clinic visits for venous phlebotomy and collection of mid-turbinate nasal swabs (MTNS). Routine vital signs, including height and weight, temperature, pulse, and oxygen saturation by pulse oximetry, were measured and recorded at each visit. At the conclusion of each visit, participants took home an MTNS kit for self-collection of a specimen at the

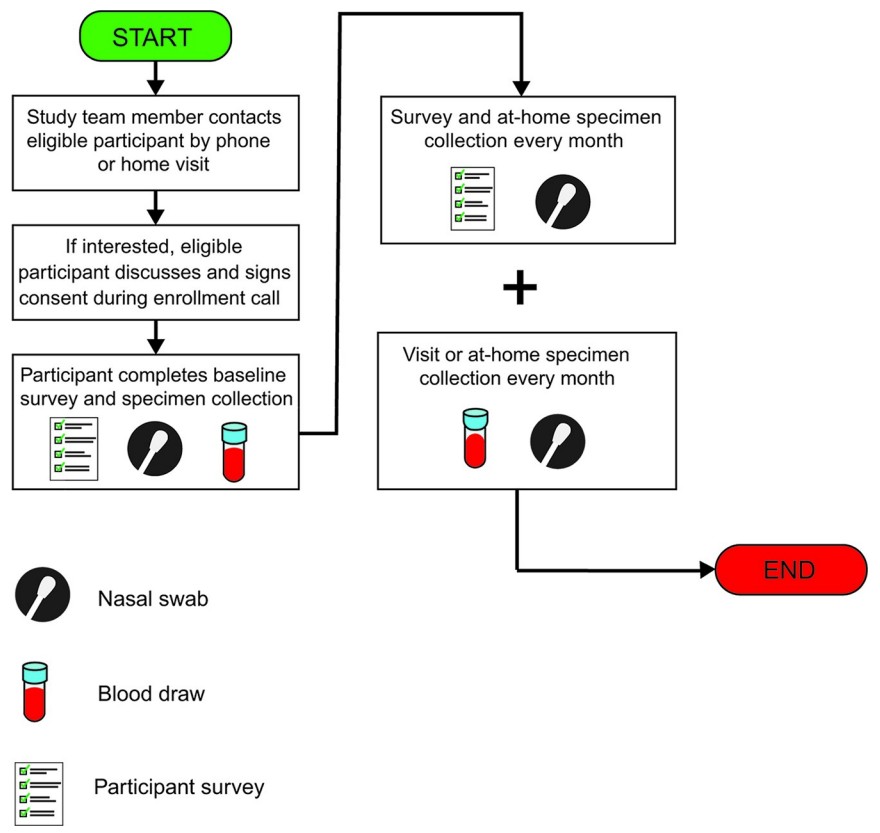

**Fig 2. Participant enrollment and study procedures.**

biweekly interval between monthly in-person visits. Participants were asked to refrigerate the nasal specimen until it could be brought to their subsequent monthly visit.

**Baseline visit.** For participants' first study visit, the take-home kit for the MTNS also included a digital thermometer. Participants were instructed to begin taking their oral temperature daily if they developed symptoms consistent with SARS-CoV-2, defined as cough, fever or chills, difficulty breathing, fatigue, body aches, diarrhea, and loss of taste or smell [22].

**Mid-Turbinate Nasal Swab (MTNS) collection.** Participants were given a nasal swab self-collection kit containing an instruction card and were also verbally instructed on usage. Participants were asked to tilt their head back at a 70-degree angle and insert the mid-turbinate swab into their right nostril until the entire tip of the swab entered the nose and resistance was felt at the turbinate. Participants then rotated the swab three times against the nasal wall and held it in place for 5 seconds. The process was repeated with the left nostril. Upon removal, the swab specimen was placed into a vial containing DNA/RNA Shield medium (Zymo Research, Irvine, CA). Vials were placed into biohazard bags and refrigerated immediately after collection.

**Venous phlebotomy.** At participants' monthly clinic visits, trained phlebotomists collected two cell preparation tubes (CPT tubes) containing up to 20 ml(cm$^3$) of venous blood in each tube (BD Vacutainer Mononuclear Cell Preparation Tube (CPT)—Sodium Citrate, BD Biosciences, San Jose, CA).

**Tasso at-home collection.** If participants opted for at-home collection only, they were mailed a self-collection kit each month consisting of a Tasso serum self-collection device (Tasso, Inc., Seattle, WA) to collect a monthly blood specimen and MTNS supplies to collect

nasal specimens every other week. For blood specimen collection, participants were instructed to wash hands and sterilize skin with an alcohol wipe. The Tasso device was adhered to skin to collect the specimen for 5 minutes, then removed. The device was then sealed into a provided biohazard bag, placed in return packaging, and shipped back to study staff for processing, along with nasal specimens.

The collection, processing, and storage of specimens followed national and international guidelines, and all processes were approved by Environmental Health Services at UNC. For full laboratory protocol, please see S3 Appendix.

## Specimen testing

**Enzyme-Linked Immunosorbent Assay (ELISA) and Abbott architect for SARS-CoV-2 antibodies.**   CPT tubes were processed for plasma up to 16 hours after collection, but typically within 8 hours of collection. Plasma collected from participants at in-person study visits and serum self-collected by participants via Tasso devices were both tested via ELISA, using the receptor binding domain (RBD) of the SARS-CoV-2 spike protein to detect total SARS-CoV-2 immunoglobulin (Ig) in plasma [23]. Specimens were also tested via the Abbott Architect Immunoassay, using the nucleocapsid protein to detect total SARS-CoV-2 IgG in plasma [24].

**Peripheral Blood Mononuclear Cell (PBMC) isolation.**   CPT tubes were further processed for PBMC collection if collected during a participant's baseline, 6-month, or 12-month study visit (See S3 Appendix) [25]. PBMCs were stored in aliquots of 3–6 million cells/µL (cells/mm$^3$) at -80˚C (193.15 K) for testing or –140˚C (133.15 K) for long-term storage and future research.

**Mid-Turbinate Nasal Swabs (MTNS).**   Following collection, MTNS were preserved in a DNA/RNA Shield medium that inactivates all infectious pathogens. MTNS collected during clinic visits were sent to the North Carolina State Laboratory for Public Health for detection of SARS-CoV-2 virus using the CDC Influenza SARS-CoV-2 (Flu SC2) multiplex polymerase chain reaction (PCR) assay [26]. MTNS collected at home by participants between clinic visits were stored at –20˚C (253.15 K) upon receipt by study staff. Using a Qiagen Viral RNA Mini Kit [27], RNA was extracted from 200µl(mm$^3$) of the DNA/RNA shield, then processed for SARS-CoV-2 PCR testing using the TaqPath™ COVID-19 Combo Kit [28].

## Data analysis

All data were stored in a HIPAA-compliant database using REDCap [29].

**Outcomes.**   Our primary outcomes of interest were the monthly incidence and cumulative prevalence of SARS-CoV-2 infection, defined as either development of SARS-CoV-2-specific antibodies as determined by ELISA (i.e., seroconversion) or clinical infection with SARS-CoV-2 confirmed by PCR testing. Secondary outcomes of interest included: (1) demographic factors (age, occupation, gender, household context, etc.), occupational factors (specific occupation, occupational exposures), and self-reported preventive behaviors (e.g. mask use, handwashing, etc.) associated with relative risk of SARS-CoV-2 infection; (2) proportion of confirmed infections that are sub-clinical and/or asymptomatic; (3) SARS-CoV-2 vaccination status or intent to vaccinate; and (4) agreement between serological testing results obtained from venous blood collection and Tasso device. Exploratory outcomes included genotypic analyses of the SARS-CoV-2 viral isolates.

**Sample size.**   A simulation study was conducted to empirically estimate the anticipated precision of the estimated prevalence of seroconversion during the study period, accounting for the study design. Data available at the time of study design suggested an infection prevalence of 5% [30]. Assuming 5% of individuals would seroconvert during the study period and

assuming correlation in infection prevalence within clusters, we estimated that the 95% confidence interval for the seroconversion estimate would have a half-width of approximately 3% based on a sample size of 300 participants.

**Proposed statistical analyses.** Sampling weights will be calculated for each participant. To account for differential nonparticipation with respect to calibration variables, base weights accounting for each household's probability of study selection will be calibrated to general Chatham County population totals derived from the American Community Survey using generalized exponential modeling [31, 32]. Calibration variables will include sampling stratum, sex, age, race/ethnicity, educational attainment, household size, and geographic region. The weighted sample is expected to be representative of adult Chatham County residents with respect to these measured calibration variables.

Weighted cross-sectional means, proportions, and categorical distributions will be estimated for survey and clinical outcomes along with accompanying standard errors and 95% confidence intervals. Specifically, monthly incidence rates and cumulative prevalence of COVID-19 will be estimated. Longitudinal models will be used to estimate changes in infection and prevalence rates over time, using generalized estimating equations with robust variance estimation to account for correlation within individuals over time and correlation within clusters. From these models, we will also evaluate demographic, socioeconomic, and geographic risk factors for COVID-19 infection.

Our longitudinal study design incorporating prospective sero-surveillance will facilitate the identification of risk factors for asymptomatic and mild cases of COVID-19 in addition to risk factors for symptomatic cases. In addition, the prospective study design will allow assessment of vaccination uptake and calculation of herd immunity estimates. All analyses will appropriately account for the complex multi-stage sample design, including weighting, stratification, and clustering with use of survey analysis software (e.g., using the R "Survey" package, SAS' survey procedures, and SUDAAN). We will also examine the spatio-temporal patterns of COVID-19 positive cases at the block group level to understand the neighborhood effects and rural-urban gradient of COVID-19 risk.

To validate serological results obtained using the Tasso at-home collection device, we will assess for concordance between ELISA antibody testing results from specimens collected by the Tasso device as compared to venous phlebotomy, which is considered the standard specimen collection procedure, through calculation of Cohen's kappa coefficient. The test performance of our ELISA antibody assay targeting the RBD protein antigen will be compared to the Abbott Architect Immunoassay for SARS-CoV-2 IgG, targeting the nucleocapsid antigen [24, 33]. In conjunction with survey-collected data on vaccination status, comparing serological assays with differing antigen targets will allow for distinction between natural immunity from SARS-CoV-2 infection and acquired immunity from vaccination and allow us to characterize and compare the resulting immune responses based on one of eight possible diagnostic combinations (Table 1).

## Results

Recruitment for this study began on August 20, 2020 and is ongoing, with enrollment anticipated to continue through July 2021. Of the 1536 eligible households sampled in Phases 1 and 2, we have received a response from 608 households (39.6%) thus far. Of these 608 households, 379 have opted out (62.3%). As of June 23, 2021, 153 participants from 152 households have been enrolled, including one participant who withdrew from the study due to moving out of the area and was subsequently replaced by another household member. Of the 153 participants who have been enrolled to date, 53 (34.6%) were enrolled from the Phase One sample and 99 (64.7%) were enrolled from the Phase Two sample. One participant not included in either

**Table 1. Distinction between natural immunity from SARS-CoV-2 infection and acquired immunity from vaccination, based on vaccination status, ELISA RBD antibody assay, and Abbott architect SARS-CoV-2 IgG assay targeting the nucleocapsid protein.**

| Possible combination | Receptor Binding Domain (RBD) to SARS-CoV-2 spike protein | SARS-CoV-2 nucleocapsid antigen | Vaccine | Possible status |
|---|---|---|---|---|
| 1 | Yes | Yes | Yes | Past infection and vaccinated |
| 2 | Yes | No | No | Likely past infection and not vaccinated |
| 3 | Yes | Yes | No | |
| 4 | No | Yes | No | |
| 5 | No | Yes | Yes | Vaccinated/uncertain immune response |
| 6 | No | No | Yes | |
| 7 | Yes | No | Yes | Vaccinated but no past infection |
| 8 | No | No | No | |

sample was also inadvertently enrolled. The majority of samples collected have successfully been analyzed; 0.8% (9 out of 1069) of nasal swabs collected for PCR testing by the North Carolina State Lab of Public Health were unusable due to shipping delays, damage to the materials, or not enough sample collected. 16.5% (20 out of 121) of samples from Tasso devices were not usable due to participant error or too small of volume of blood collected, or other causes.

## Discussion

Our study design has several unique and novel aspects that maximize its potential for far-reaching impact. Foremost among these is the probability-based sampling design, which enables us to obtain a more complete picture of how and why SARS-CoV-2 may be spreading throughout rural, under-resourced communities. This sampling design helps to ensure that the estimates generated are generalizable to the household population in Chatham County, which is demographically similar to the state of North Carolina (S1 Table). As a result, this data can be used to inform planning and policy decisions that benefit the state as a whole.

Another advantage of our approach is the inclusion of lower-income communities and rural areas, which, despite the initial focus on urban areas, have been particularly stricken by the COVID-19 pandemic in North Carolina [34]. Approximately 40 percent of North Carolina's population resides in rural communities like Chatham County [35]. Rural communities often have fewer clinics and hospitals, and therefore less access to testing and treatment. Relying exclusively on positive tests and hospital admissions has the potential to miss a substantial proportion of infections in these communities. In addition, rural communities host major agricultural production facilities with many workers deemed "essential" [36]. Such facilities have been particularly hard hit by outbreaks of SARS-CoV-2 infection [1]. Conducting our study within a county that hosts multiple meat-processing plants gives us the opportunity to study how agricultural facilities affect community transmission of SARS-CoV-2.

Our study cohort is also exceptionally well-characterized for multiple reasons: 1) building our cohort on a pre-existing population-based cohort expedited enrollment and data collection early in the pandemic, enabling us to collect data from the same individuals over an extended period of time and 2) collecting information of substantial depth and breadth regarding participants' clinical symptoms of and occupational exposures to SARS-CoV-2, infection prevention behaviors, mental health, and perceptions of the epidemic through repeated electronic questionnaires. The timing and frequency of survey data collection allows us to examine shifts in participants' beliefs and behaviors before SARS-CoV-2 vaccines were available, during vaccine roll-out, and after widespread vaccination. Due to the demographic similarity of Chatham County to the state of North Carolina, this information can inform vaccine planning

efforts at the community and state levels. The frequency of specimen collection also allows us to quickly detect emerging trends, including surges in infections, vaccination uptake, and vaccine effectiveness.

While our approach has many strengths, it also has some limitations. Due to differential rates of participation, our cohort may overrepresent older age groups and persons more likely to be at home during the pandemic. It may also underestimate those with pre-existing conditions or other high-risk individuals due to a reluctance to attend in-person clinic visits. Furthermore, our sampling frame does not cover institutionalized individuals, including persons residing in nursing homes and individuals who are incarcerated. We attempted to minimize potential biases by offering at-home collection kits, implementing multiple modes of recruitment over the span of several months, including home visits, telephone calls, and electronic and postal mailings, and using weight calibration adjustments during analysis to incorporate factors associated with study participation. In addition to minimizing potential biases, our adaptive approach to specimen collection has the advantage of being able to capture specimens from participants with active infections who otherwise would have been restricted from attending clinic visits in person. However, despite the opportunity to conduct at-home specimen collection as needed, our sampling strategy may not capture all incident infections occurring between nasal swab collections because of the relatively short duration of detectable viral shedding. Finally, our study will identify asymptomatic and undetected infections through monthly serological testing, although this could create more variability in estimates of infection onset.

## Conclusions

This article describes the protocol for a prospective, longitudinal, population-based cohort study that will generate crucial data about the prevalence and incidence of SARS-CoV-2 infection, trends in SARS-CoV-2 infection over time, vaccination uptake, and risk factors for SARS-CoV-2 infection within a semi-urban and rural county in North Carolina, with broad application for informing state public health policy. This study design is adaptable to different regional settings, and therefore can be duplicated to provide epidemiological knowledge across a larger geographic area. We know of two other groups that have modeled their studies after ours thus far, covering the areas of Pitt County in eastern North Carolina (https://compactstudy.ecu.edu/) and Cabarrus County in south-central North Carolina (https://murdock-study.com/ongoing-studies/murdock-cabarrus-county-covid-19-prevalence-and-immunity-study/), and we have shared study protocols, best practices, and our surveys with these groups. Such collaborative research networks provide opportunities to aggregate data for comparison and offer surveillance infrastructure urgently needed for future pandemic preparedness.

## Supporting information

**S1 Table. Demographic characteristics of Chatham County residents as compared to North Carolina residents.** Sources: Percent of households under federal poverty rate are from 2019 U.S. Census Small Area Income and Poverty Estimates (SAIPE). Race and ethnicity are from 2019 American Community Survey (ACS) 5-year estimates, whereas all other demographics use 2019 ACS 1-year estimates.
(DOCX)

**S1 Fig. Map of North Carolina counties highlighting Chatham County.**
(TIF)

**S2 Fig. Informational postcard describing study.**
(TIF)

**S1 Appendix. Baseline participant questionnaire.**
(PDF)

**S2 Appendix. Biweekly participant survey.**
(PDF)

**S3 Appendix. Laboratory Standard Operating Procedures (SOPs).**
(DOCX)

## Acknowledgments

The authors thank the participants for their willingness to contribute to advancing our understanding of the SARS-CoV-2 epidemic and its impact on rural communities, especially during the early and uncertain months of the pandemic. We also acknowledge Paul N. Zivich, PhD for the creation of the schematic depicting participant enrollment and study procedures. We owe special thanks to Meredith Bazemore, Director of the Office of Rural Initiatives at UNC, for her aid in community engagement.

## Author Contributions

**Conceptualization:** John Wallace, Michael Zelek, Jessie K. Edwards, Aaron T. Fleischauer, Emily J. Ciccone, Bonnie E. Shook-Sa, Allison E. Aiello, Ross M. Boyce.

**Data curation:** Elyse M. Miller, Rawan Ajeen, Jaclyn Karasik, Carmen Mendoza, Khou Xiong.

**Formal analysis:** Elyse M. Miller, Elle A. Law, Rawan Ajeen, Carmen Mendoza.

**Funding acquisition:** Aaron T. Fleischauer, Allison E. Aiello, Ross M. Boyce.

**Investigation:** Haley Abernathy, Haley Garrett, Elise King.

**Methodology:** Elyse M. Miller, Elle A. Law, Jaclyn Karasik, Carmen Mendoza, Cherese Beatty, Emily J. Ciccone, Bonnie E. Shook-Sa, Allison E. Aiello, Ross M. Boyce.

**Project administration:** Khou Xiong, Cherese Beatty, Emily J. Ciccone, Allison E. Aiello, Ross M. Boyce.

**Resources:** Allison E. Aiello, Ross M. Boyce.

**Software:** Rawan Ajeen.

**Supervision:** Emily J. Ciccone, Allison E. Aiello, Ross M. Boyce.

**Visualization:** Elyse M. Miller, Elle A. Law.

**Writing – original draft:** Elyse M. Miller, Elle A. Law, Jaclyn Karasik, Haley Abernathy, Haley Garrett, Elise King, Bonnie E. Shook-Sa.

**Writing – review & editing:** Elyse M. Miller, Elle A. Law, Rawan Ajeen, Jaclyn Karasik, Carmen Mendoza, Haley Abernathy, Haley Garrett, Elise King, John Wallace, Michael Zelek, Jessie K. Edwards, Khou Xiong, Cherese Beatty, Aaron T. Fleischauer, Emily J. Ciccone, Bonnie E. Shook-Sa, Allison E. Aiello, Ross M. Boyce.

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
