## [Decision Letter · Decision Letter 0]

27 Aug 2021

PONE-D-21-21567

SARS-CoV-2 infection in central North Carolina: Protocol for a population-based longitudinal cohort study

PLOS ONE

Dear Dr. Boyce,

Thank you for submitting your manuscript to PLOS ONE. After careful consideration, we feel that it has merit but does not fully meet PLOS ONE’s publication criteria as it currently stands. Therefore, we invite you to submit a revised version of the manuscript that addresses the points raised during the review process.

The reviewers recommend that you make minor revisions to the manuscript. Please attend to their concerns and then return the revised manuscript as advised in this letter.

We look forward to receiving your revised manuscript.

Kind regards,

Martin Chtolongo Simuunza, PhD

Academic Editor

PLOS ONE

Journal Requirements:

2.Please review your reference list to ensure that it is complete and correct. If you have cited papers that have been retracted, please include the rationale for doing so in the manuscript text, or remove these references and replace them with relevant current references. Any changes to the reference list should be mentioned in the rebuttal letter that accompanies your revised manuscript. If you need to cite a retracted article, indicate the article’s retracted status in the References list and also include a citation and full reference for the retraction notice.

Reviewers' comments:

Reviewer's Responses to Questions

**Comments to the Author**

1. Does the manuscript provide a valid rationale for the proposed study, with clearly identified and justified research questions?

Reviewer #1: Yes

Reviewer #2: Yes

2. Is the protocol technically sound and planned in a manner that will lead to a meaningful outcome and allow testing the stated hypotheses?

Reviewer #1: Yes

Reviewer #2: Yes

3. Is the methodology feasible and described in sufficient detail to allow the work to be replicable?

Reviewer #1: Yes

Reviewer #2: Yes

4. Have the authors described where all data underlying the findings will be made available when the study is complete?

Reviewer #1: Yes

Reviewer #2: No

5. Is the manuscript presented in an intelligible fashion and written in standard English?

Reviewer #1: Yes

Reviewer #2: Yes

6. Review Comments to the Author

You may also provide optional suggestions and comments to authors that they might find helpful in planning their study.

Reviewer #1: The authors of this work try to generate accurate population-based estimates of SARS-CoV-2 infection in the North Carolina and specifically in Chatham County. The protocol is comprehensive and many parameter and challenges were taken to obtain proper samples for the study. Although there initial target was 300 participants but they managed to obtain 153. The protocol was clearly written and easy to follow and comprehend. However, not presenting the results of the study make it difficult to judge if the protocol and the samples were collected in a good condition to be examined or not. Despite of that, the protocol was thoroughly described and it is important to be shared with the scientific community.

Minor comments:

- Sentences from 109-111 and 116-118 are redundant. Remove one of them.

- S1 figure has no label.

- Figure 2 resolution is not good.

- Page 251 and 259 µl not ul. Change it throughout the manuscript if there are more.

- It is not clear why there are multiple rows in Table 1.

Reviewer #2: Great work! Sounds like great writeup of methodology and preliminary sample results. Please see comments to improve and emphasize this is a preliminary report in the title and in the text early on that the study is ongoing so the reader is crystal clear on that. Otherwise thought it was a nice read. I have wondered what the difference in what the true difference between hospital-reported and "actual" counts in the community. Topic very nicely selected and addressed.

7. PLOS authors have the option to publish the peer review history of their article (what does this mean?). If published, this will include your full peer review and any attached files.

Reviewer #1: No

Reviewer #2: No

---

## [Decision Letter · Decision Letter 1]

12 Oct 2021

SARS-CoV-2 infection in central North Carolina: Protocol for a population-based longitudinal cohort study and preliminary participant results

PONE-D-21-21567R1

Dear Dr. Boyce,

We’re pleased to inform you that your manuscript has been judged scientifically suitable for publication and will be formally accepted for publication once it meets all outstanding technical requirements.

Kind regards,

Martin Chtolongo Simuunza, PhD

Academic Editor

PLOS ONE

Additional Editor Comments (optional):

Reviewers' comments:

Reviewer's Responses to Questions

**Comments to the Author**

1. Does the manuscript provide a valid rationale for the proposed study, with clearly identified and justified research questions?

Reviewer #1: Yes

Reviewer #2: Yes

2. Is the protocol technically sound and planned in a manner that will lead to a meaningful outcome and allow testing the stated hypotheses?

Reviewer #1: Yes

Reviewer #2: Yes

3. Is the methodology feasible and described in sufficient detail to allow the work to be replicable?

Reviewer #1: Yes

Reviewer #2: Yes

4. Have the authors described where all data underlying the findings will be made available when the study is complete?

Reviewer #1: Yes

Reviewer #2: No

5. Is the manuscript presented in an intelligible fashion and written in standard English?

Reviewer #1: Yes

Reviewer #2: Yes

6. Review Comments to the Author

You may also provide optional suggestions and comments to authors that they might find helpful in planning their study.

Reviewer #1: Authors have satisfactorily replied to all the comments except for S1 figure, which I think will be more clearer to add a title and a caption to it.

Reviewer #2: Journal and/or editor need to comment on if availability via request only is sufficiently "public" access to your study's data. That is why I answered no. There is no guarantee ALL requests for your data will be approved and you state someone is going to be an investigator that requests it. Maybe, maybe not. May still be a member of the public. And who qualifies as an investigator may be defined differently from one institution to the next.

7. PLOS authors have the option to publish the peer review history of their article (what does this mean?). If published, this will include your full peer review and any attached files.

Reviewer #1: No

Reviewer #2: No

---

## [Editor Report · Acceptance letter]

14 Oct 2021

PONE-D-21-21567R1 

SARS-CoV-2 infection in central North Carolina: Protocol for a population-based longitudinal cohort study and preliminary participant results 

Dear Dr. Boyce:

I'm pleased to inform you that your manuscript has been deemed suitable for publication in PLOS ONE. Congratulations! Your manuscript is now with our production department. 

Kind regards, 

on behalf of

Dr. Martin Chtolongo Simuunza 

Academic Editor

PLOS ONE